

# The effectiveness of the Peyton's 4-step teaching approach on skill acquisition of procedures in health professions education: A systematic review and meta-analysis with integrated meta-regression

Katia Giacomino[*], Rahel Caliesch and Karl Martin Sattelmayer[*]

School of Health Sciences, HES-SO Valais-Wallis, Leukerbad, Switzerland
[*] These authors contributed equally to this work.

Corresponding author
Karl Martin Sattelmayer,
martin.sattelmayer@hevs.ch

## ABSTRACT

**Background**. Acquisition of procedures is an important element in health professions education. Traditionally procedures are taught using a "see one - do one" approach. That is a teacher demonstrates and describes a procedure and afterwards the students practice the procedure. A more recent teaching approach for the acquisition of procedural skills was presented by Walker and Peyton. Peyton's teaching approach is a stepwise teaching approach and consists of the following four steps: demonstration, deconstruction, comprehension and performance. The aims of this study were (i) to systematically evaluate the effectiveness of Peyton's 4–step teaching approach on the acquisition of procedural skills in health professions education and (ii) to evaluate whether studies with fewer students per teacher showed a larger between group difference than studies with more students per teacher.

**Methods**. We searched in Medline, PsycInfo, Embase and ERIC for eligible studies. Records were screened by two independent reviewers. A random effects meta-analysis was performed to evaluate skill acquisition and time needed to perform the procedures at post-acquisition and retention tests. A meta-regression was used to explore the effect of the number of students per teacher on the estimated effect of the educational interventions.

**Results**. An effect size of 0.45 SMD (95% CI [0.15; 0.75]) at post-acquisition and 0.7 SMD (95% CI [−0.09; 1.49]) at retention testing were in favour of Peyton's teaching approach for skill acquisition. The groups using Peyton's teaching approach needed considerably less time to perform the procedure at post-acquisition (SMD: −0.8; 95% [CI −2.13 to 1.62]) and retention (SMD: −2.65; 95% CI [−7.77 to 2.47]) testing. The effectiveness of Peyton's teaching approach was less clear in subgroup analyses using peer teachers. Meta-regression showed that the number of students per teacher was an important moderator variable.

**Conclusion**. Peyton's teaching approach is an effective teaching approach for skill acquisition of procedural skills in health professions education. When peer students or student tutors are used as teachers the effectiveness of Peyton's teaching approach is less clear. Peyton's teaching approach is more effective when small groups with few students per teacher are used.

## INTRODUCTION

Acquisition of procedures is an important element in health professions education (*Grantcharov & Reznick, 2008*). Historically, the study of the acquisition of procedural skills was primarily in the field of medical and especially surgical education. However, other health professions such as nursing and physiotherapy education have developed assessment and teaching approaches for these skills as well (*Oermann, Muckler & Morgan, 2016*; *Sattelmayer, Hilfiker & Baer, 2017*). Defining procedural skills is challenging. *Michels, Evans & Blok (2012)* reported that there is considerable overlap between the terms clinical skills, psychomotor skills and procedural skills.

Traditionally procedures are taught using a "see one - do one" approach. This means that a teacher demonstrates and describes a procedure and afterwards the students are asked to practice the procedure. This is referred to as Halsted's teaching approach, which is based on the surgeon *Halsted (1904)*. The approach was used as an element to redesign surgical education and create a new system for training young surgeons (*Cameron, 1997*). Although the "see one - do one" approach is often used in the training of health professionals, there is criticism of this approach. First, the approach has been used for decades and does not adhere to recent principles of adult learning such as active learner involvement (*McLeod et al., 2001*). Furthermore, it was reported that patient safety might be at risk because complex procedures cannot be acquired after a single observation and practice trial (*Kotsis & Chung, 2013*). Given the diversity of existing procedures today, others argue that the teaching approach should be modified to "see many, learn from the result and do many" (*Rohrich, 2006*).

A more recent teaching approach for the acquisition of procedural skills was presented by *Walker & Peyton (1998)*. Peyton's teaching approach is a stepwise teaching approach and consists of the following four steps: (i) step 1 refers to the demonstration of the whole procedure in real time ("demonstration"); (ii) in step 2 the teacher repeats the demonstration but this time all procedural sub-steps are described ("deconstruction"); (iii) during step 3 the student talks the teacher through the procedure. The teacher performs the procedure under the guidance of the student ("comprehension") and (iv) in step 4 the students carry out the procedure on their own initiative ("performance").

A similar stepwise teaching approach was presented by George (*American College of Surgeons, 1997*) and later published by *George & Doto (2001)*. Originally, it was developed as an educational technique to support the American College of Surgeon's Advanced Trauma Life Support course. In contrast to Peyton's teaching approach, George and Doto used five steps. Within Peyton's teaching approach, two of the five steps are collated into a single step. George and Doto based their teaching approach on Simpson's taxonomy of the psychomotor domain (*Simpson, 1966*).

Especially the third step seems to be important in Peyton's teaching approach and was assumed to be beneficial for skill acquisition. The process of guiding the teacher through the procedure requires the student to remember and think about the first two steps before giving the teacher the necessary information (*Gradl-Dietsch et al., 2016*). This process could help students to organise their thoughts and support student-centred learning (*Lom, 2012*). Similarly, Balafoutas and colleagues (*2019*) argue that students need to manipulate the information stored in their working memory based on the information provided in the first two steps. This could support the transfer of relevant information into the long-term memory. Other authors have argued that recognising the effects of the instructions on the performance could be a valuable source of feedback and might improve metacognitive skills (*Herrmann-Werner et al., 2013*). In addition, *Rossettini et al. (2017)* mentioned that Peyton's third step involves elements of mental practice. That is, the students have the possibility to develop a mental representation of the movement in absence of an active movement. There exists evidence that mental practice is effective for skill acquisition of procedures in health professions education (*Sattelmayer et al., 2016*).

Besides the third step, the fourth step is also of educational importance as in this step the teacher provides feedback to the learner. A systematic review by Issenberg reported that the opportunity to provide feedback is a key component for effective skill acquisition in simulation-based medical education (*Issenberg et al., 2005*). In addition, the fourth step is also supported by Bandura's scaffolding theory (*Schunk, 2012*).

One of the strengths of Peyton's teaching approach is that it can be effectively combined with other instructional design strategies, which allows the simultaneous delivery of theoretical concepts along with complex procedural skills. For example, *Tambi et al. (2018)* combined Gagne's instructional model (*Gagne et al., 2005*) with Peyton's teaching approach to design a bioinformatics lesson plan for medical students and *Ng (2014)* combined both teaching approaches for slit-lamp teaching.

However, one could assume that the step-by-step approach would require considerably more time for teaching. The traditional teaching approach consist typically of two steps (demonstration and practice). The additional two steps might be assumed to be time-consuming. However, in contrast to this, several authors have reported that not more time was required using Peyton's approach (*Krautter et al., 2011*; *Rossettini et al., 2017*).

Several randomised controlled trials have evaluated the effectiveness of Peyton's teaching approach. The results of these studies are not always consistent. Some trials have reported findings in favour of Peyton's approach (e.g., *Balafoutas et al., 2019*; *Rossettini et al., 2017*). *Rossettini et al. (2017)* showed that acquisition of a cervical mobilisation technique was considerable higher in the Peyton group compared to a standard teaching group. In contrast, *Orde, Celenza & Pinder (2010)* have reported that Peyton's teaching approach showed only minor differences on skill acquisition regarding insertion of a laryngeal mask airway at post-acquisition and retention testing compared to a traditional teaching approach.

Originally Peyton's teaching approach was designed for a student-teacher ratio of 1:1 (*Nikendei et al., 2014*). However, such a ratio is difficult to achieve in educational

institutions. Therefore, from a pragmatic point of view it is important to evaluate whether Peytons's teaching approach can be used with more students per teacher.

These inconsistencies should be further investigated through a systematic review. Therefore, the aims of this study were (i) to systematically evaluate the effectiveness of Peyton's 4–step teaching approach on the acquisition of procedural skills in health professions education and (ii) to evaluate whether studies with fewer students per teacher (i.e., the student-teacher ratio) showed a larger between group difference than studies with more students per teacher.

## MATERIALS & METHODS

A protocol of this systematic review was registered in the OSF registries: https://doi.org/10.17605/OSF.IO/5UE7C. To improve clarity of reporting the PRISMA statement was followed (*Liberati et al., 2009*).

### Searches

We searched the following electronic databases for eligible studies: Medline, PsycInfo, Embase and Education Resources Information Center (ERIC). The search was performed by KMS. No restrictions regarding recency or publication language were set. The search strategy was prepared using two blocks. The first block consisted of terms relevant for the identification of the population (i.e., students in health professions education). We searched for keywords and mapped the keywords to relevant subject headings. The second block was designed to identify studies using Peyton's teaching approach. Both search blocks were combined using the Boolean operator "and". The search strategy is reported in Appendix S1. In addition, references of included studies were checked for potential eligible studies.

### Selection criteria

The following selection criteria were applied.

#### *Types of studies to be included*

Randomised controlled trials were included. If sufficient data were available cross-over studies were eligible as well.

#### *Participants*

Only studies reporting on students in health professions education were included. Health professions education was used as an umbrella term for medical and allied health profession education (e.g., physiotherapy or nursing education). We included studies reporting on undergraduate and postgraduate students.

#### *Interventions*

Studies needed to investigate Peyton's 4-step approach for inclusion in at least one study arm (i.e., all 4 steps were used together).

### *Comparator*

Studies needed to have a comparator group. The comparator could be a specific educational intervention (e.g., team-based education or peer teaching), educational practice as usual (e.g., a "see one - do one") or a sham intervention.

### *Outcomes*

The primary outcome for this review was the evaluation of procedural skills. These could be evaluated using a performance metric such as a procedure specific checklist or a global rating scale. To be included studies had to report on this outcome. The secondary outcome was the time needed to perform the procedure. If multiple procedures were trained one procedure was selected for inclusion in order to avoid a unit of analysis issue (i.e., in order to avoid including the same participants twice within a single analysis). Means and standard deviations of continuous outcomes were extracted. If standard deviations were not reported we imputed standard deviations based on standard errors or confidence intervals as suggested in the Cochrane Handbook (*Higgins et al., 2019*).

## Study selection and data extraction

Records were screened by two independent reviewers (RC and KMS). The screening procedure was performed using the Rayyan software (*Ouzzani et al., 2016*). Disagreements were solved by discussion between RC and KMS. If a referee was needed KG was consulted. One reviewer (KMS) extracted relevant data into an electronic database and a second reviewer (KG) controlled the data.

## Risk of bias assessment

The risk of bias was evaluated using the Cochrane risk of bias tool (*Higgins et al., 2011*). A human reviewer (KMS) evaluated all included studies with respect to these items: sequence generation, allocation concealment, blinding of (a) participants and personnel and (b) outcome assessors, incomplete outcome data and selective reporting. Evaluations were compared against a machine learning classification of the risk of bias with the application "RobotReviewer" (*Marshall, Kuiper & Wallace, 2015*). Disagreements were solved by discussion with a third person.

## Strategy for data synthesis

The primary endpoint for evaluating the effectiveness of the comparisons was at the end of the intervention. A secondary analysis was performed using data from the longest available follow up endpoint.

## Data analysis

The analysis was performed using the statistical software package R (*R Core Team, 2019*). A meta-analysis of pairwise comparisons was performed using the meta package (*Schwarzer, 2007*). A random effects model was used for the analysis and effectiveness was reported using standardized effect sizes (Hedges' g) and corresponding 95% confidence intervals. The Hartung, Knapp, Sidik, Jonkmann adjustment was applied to achieve robust estimations of the treatment effect (*InTHout, Ioannidis & Borm, 2014*). Effect sizes were interpreted following *Cohen (1992)*. This means that an effect size of 0.2 was considered as small, 0.5

as medium and 0.8 as large. Statistical heterogeneity was assessed with $I^2$ statistics using the guidelines presented in the Cochrane handbook for systematic reviews of interventions (*Higgins & Green, 2011*). The following categories were applied: 0–40% might not be important, 30–60% moderate heterogeneity, 50–90% substantial heterogeneity and 75–100% considerable heterogeneity.

A mixed effects meta-regression was performed using the meta package (*Schwarzer, 2007*). We explored the effect of the students per teacher on the estimated effect of the educational interventions. The number of students per teacher during the procedural skills training was used as moderator variable.

## RESULTS

### Findings of the search

The electronic search on the databases Medline, PsycInfo, Embase and ERIC identified 482 potential eligible records. In addition, the screening of the abstracts identified 5 further records. After removing 45 duplicates, 442 titles and abstracts were screened. In this phase of the selection process 405 records were excluded. The full-texts of the remaining 37 records were assessed for eligibility and 23 records were excluded with the following reasons: 12 records reported an intervention, which was not eligible for inclusion (*Bode et al., 2012*; *Bube, Konge & Hansen, 2017*; *Craven et al., 2018*; *Custers et al., 1999*; *Handley & Handley, 1998*; *Hill et al., 2010*; *Holmes et al., 1998*; *Krautter et al., 2015*; *Liu & Hunt, 2017*; *Velmahos et al., 2004*; *Wirth et al., 2018*; *Yoganathan et al., 2018*); 8 records used a study design, which was not eligible for inclusion (*Easton, Stratford-Martin & Atherton, 2012*; *Mishra & Dornan, 2003*; *Nikendei et al., 2014*; *Schroder et al., 2017*; *Skrzypek et al., 2018*; *Smith et al., 2019*; *Sopka et al., 2012*; *Tommaso, 2016*); 2 records were excluded because of missing data (*Archer, Van Hoving & De Villiers, 2015*; *Seymour-Walsh et al., 2015*) and 1 record did not use the specified primary outcome assessment for procedural skills (*Greif et al., 2010*). Finally, 14 studies were included into this systematic review. An overview of the selection process is presented in Fig. 1. During the study selection process, 6 conflicts occurred, representing 1.4% of the total decisions.

### Included studies

The 14 included studies in this review were all randomised controlled studies. An overview of included studies and study characteristics is presented Table 1. Most of the included studies were conducted in Germany ($n = 10$). Four studies with 3 or 4 study arms were included (*Gradl-Dietsch et al., 2018*; *Herrmann-Werner et al., 2013*; *Münster et al., 2016*; *Ruesseler et al., 2019*). In these cases, study arms investigating Peyton's teaching approach or a standard teaching approach were included. Study arms using an intervention not eligible for inclusion were excluded from this review. For example, *Gradl-Dietsch et al. (2018)* reported 4 study arms. The study arms peer teaching and peer teaching using Peyton's teaching approach were included. Not included were the study arms team-based learning and video-based learning. All used study arms are presented in Table 1. The included participants in most studies were within medical education. A range from first year medical students to residents in obstetrics and gynaecology was identified. Two studies
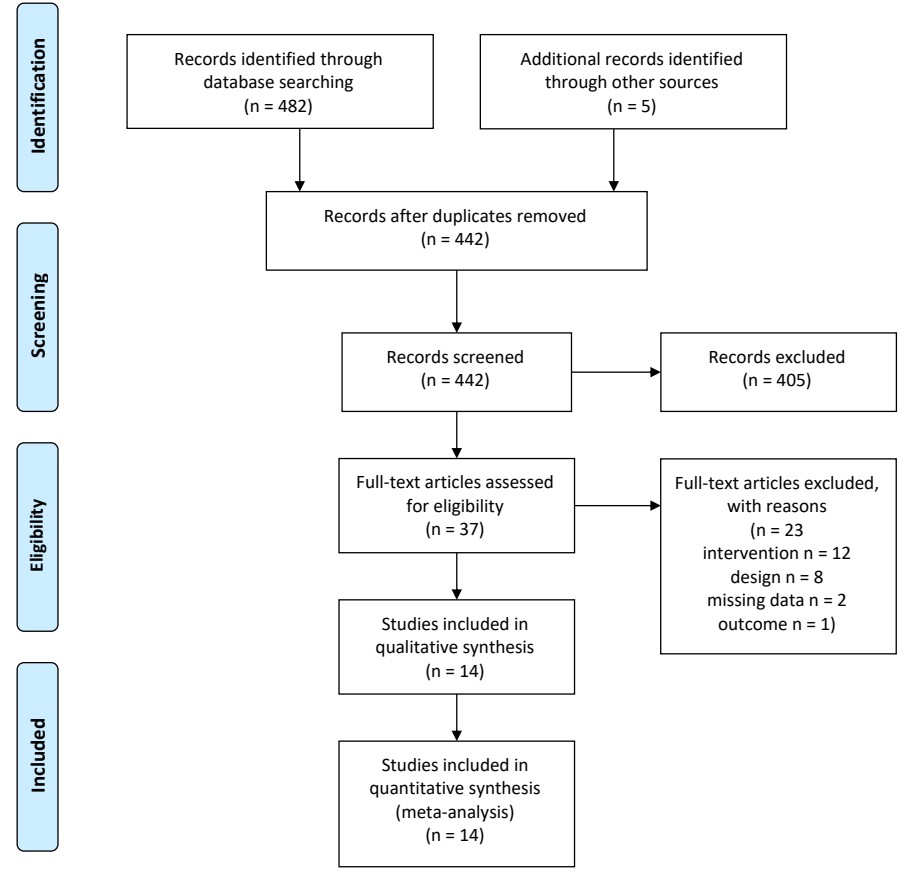

**Figure 1** Prisma flow diagram.

used participants from nursing education (*Lapucci et al., 2018*; *Orde, Celenza & Pinder, 2010*) and one study was conducted with participants from physiotherapy education (*Rossettini et al., 2017*). A broad range of trained procedures has been identified. For example, basic surgical skills (*Ruesseler et al., 2019*), spine mobilisations (*Gradl-Dietsch et al., 2016*; *Rossettini et al., 2017*), musculoskeletal ultrasound (*Gradl-Dietsch et al., 2019*) or cardiopulmonary resuscitation (*Jenko, Frangež & Manohin, 2012*) were used as procedures. Several modified versions of Peyton's teaching approaches were used in the experimental groups. All studies with exception of five studies (*Gradl-Dietsch et al., 2019*; *Gradl-Dietsch et al., 2018*; *Herrmann-Werner et al., 2013*; *Münster et al., 2016*; *Ruesseler et al., 2019*) used a standard version of Peytons's teaching approach.

The study of *Herrmann-Werner et al. (2013)* used a best practice skills laboratory, which consisted of structured individual feedback, performance on manikins and Peyton's teaching approach supervised by student tutors. Three studies (*Gradl-Dietsch et al., 2019*; *Gradl-Dietsch et al., 2018*; *Münster et al., 2016*) used peer or student teachers for the teaching events and *Ruesseler et al. (2019)* used a video 4 -step approach.

The teaching approach in the control groups was described as traditional Halsted teaching (*Balafoutas et al., 2019*; *Romero et al., 2018*), peer teaching or student tutors

**Table 1 Characteristics of included studies.**

| Study | Design/ Country | Participants | Procedure trained[a] | Teaching approach in experimental group | Teaching approach in control group | Time required for teaching | Student teacher ratio | Outcome measurements[a] | Endpoints |
|---|---|---|---|---|---|---|---|---|---|
| *Balafoutas et al. (2019)* | RCT/ Germany | n = 16 residents in obstetrics and gynaecology | Laparoscopic suturing and knot-tying training | Deconstruction of teaching practical clinical skills in 4 steps (demonstration, deconstruction, comprehension, execution) | Traditional Halsted teaching (demonstration followed by execution) | Instructions in both groups had a duration of 30 min. Afterwards the groups received an equal amount of time for practice | 1:1 | Objective Structured Assessment of Technical Skills tool; number of correct knots; mean time required for knot tying | Post-test (after the training) |
| *Gradl-Dietsch et al. (2019)* | Randomised cross over study/ Germany | n = 491 second year medical students | Musculoskeletal ultrasound (shoulder and knee joint) | Peer teaching according to the Peyton method (demonstration, deconstruction, comprehension, execution) | Peer teaching (demonstration and execution) | A lesson lasted 75 min (15 min theory, 15 min demonstration, 45 min training) in both groups | 9:1 | Objective structured practical examination; binary performance checklist; global rating scale; time required | Post-test 2 weeks after training |
| *Gradl-Dietsch et al. (2016)* | RCT/ Germany | n = 95 third to fifth year medical students | Manual therapy and specific manipulative and diagnostic techniques for the spine | Instructions following the approach of Peyton. Steps 1 and 2 within group. Steps 3 and 4 individually (demonstrate, talk the trainee through, trainee talks trainer through, trainee does) | Standard instructions (demonstration and practice) | Session duration was 120 min (30 min theory and 60 min training) in both groups | 1:1 | Objective Structured Practical Examination; binary performance checklist; Multiple choice exam (principles of manual therapy) | Post-test (4 weeks after training), retention test (6 month) |
| *Gradl-Dietsch et al. (2018)* | 4-arm RCT (the arms peer teaching and Peyton peer teaching were included)/ Germany | n = 38 s year medical students | Echocardiography including technical requirements and patient preparation | Peer teachers demonstrated according to Peyton's approach (demonstrate, talk the trainee through, trainee talks trainer through, trainee does) | Peer teaching (peer teachers demonstrated the procedure; students then practised the skills on each other) | Session duration was 90 min in all groups | 3:1 for peer Peyton; n.a. for peer teaching | Objective structured practical examination; binary performance checklist; global rating scale; multiple choice test | Post-test (2 weeks after the training) |
| *Herrmann-Werner et al. (2013)* | 4 arm RCT/ Germany | n = 94 undergraduate medical students | Nasogastral tube insertion and intravenous cannulation | Student tutors supervised a best practice skills laboratory training consisting of structured individual feedback, performance on manikins and Peyton's "Four-Step-Approach (demonstration, deconstruction, comprehension, performance) | Student tutors supervised a "see one, do one", teaching | The length of teaching sessions did not significantly differ between groups | 3:1 | Video recordings of performances were evaluated with binary and global checklists; amount of time needed | Post-test (immediately after training) and retention test (6 months after the training) |
| *Jenko, Frangež & Manohin (2012)* | RCT/ Slovenia | N = 126 first-year medical students | Cardiopulmonary resuscitation | Peyton's 4 stage approach (demonstration, deconstruction, formulation, performance) | 2-stage approach (demonstration slow speed and commentary followed by performance) | The duration of the course was 4.5 h for both groups | 12:1 | Performance scores measured with the manikin: compression depth, rate and hand placement | Post-test (immediately after training) |

| Study | Design/ Country | Participants | Procedure trained[a] | Teaching approach in experimental group | Teaching approach in control group | Time required for teaching | Student teacher ratio | Outcome measurements[a] | Endpoints |
|---|---|---|---|---|---|---|---|---|---|
| *Krautter et al. (2011)* | RCT/ Germany | n = 34 second- and third-year medical students | Gastric-tube insertion using a manikin | Peyton's Four-Step Approach (demonstrate, talk the trainee through, trainee talks trainer through, trainee does) | Standard instructions: consisting of demonstration with detailed commentary and time to ask questions | No difference between length of instructions between groups | 1:1 | Acceptance ratings, length of time for instructions, lengths of time for first independent performance, video ratings of performance including ( binary checklist and global rating scale) | Post-test |
| *Lapucci et al. (2018)* | RCT/ Italy | n = 60 first- and second-year nursing students | Cardio-Pulmonary Re-animation | Peyton's 4-step teaching method (demonstrate, deconstruction, comprehension, execution) | 2 step method described by Orde (Peyton's step 2 and step 4). | Both groups received 15 min of training | 10:1 | Performance scores: insuf-ficient chest compressions, excessive chest compressions, effective chest compressions and effective ventilations | Post-test (after training) |
| *Lund et al. (2012)* | RCT/ Germany | n = 84 first-year medical students | Intravenous cannulation on a part-task-trainer model in the shape of a human arm | Training in a skills lab using Pey-ton's 4 step approach | Traditional bedside teaching based on "see one, do one". | Length of teach-ing sessions was similar between groups | 3:1 | Video rating with binary checklist, global rating scale, time needed and number of attempts and patient ratings | Post-test in clin-ical setting with volunteer stu-dents. |
| *Münster et al. (2016)* | 3-arm RCT (the arms Peyton and standard teaching were included)/ Ger-many | n = 103 second- and third-semester medi-cal students | Cardiopulmonary resuscitation | Student tutors used Peyton's 4 step approach (demonstration, deconstruction, modified step comprehension for groups, exe-cution) | Student tutors used a standard teaching method: Peyton's step 2 and 4 (deconstruction and per-formance steps) | The practical instructions had a duration of 90 min | median group size 13 | Binary performance checklist and performance data of the resuscitation phantom | Post-test (1 week after train-ing), retention test 5-6 month after training) |
| *Orde, Celenza & Pinder (2010)* | RCT/ Australia | n = 120 final year medical students, nurses and student nurses | Insertion of a Laryngeal Mask Airway on an airway training manikin | 4-stage teaching (demonstration, deconstruction, formulation, performance) | 2-stage teaching (deconstruction and performance steps) | n.a. | n.a. | Time taken for insertion, num-ber of steps correctly and incorrectly performed, and number of steps omitted | Post-test (im-mediately after training), re-tention test (2 months after training) |
| *Romero et al. (2018)* | RCT/ Germany | n = 60 third- to sixth-year medi-cal students | Intracorporal suturing and knot tying | Peyton's Four-Step approach (demonstration, deconstruction, comprehension, performance) | Halsted teaching; the teacher demonstrated once afterwards the students practiced on their own | Standardised training time of 60 min in both groups | 1:1 | Objective Structured Assessment of Technical Skills (OSATS) with checklist and global rating scale, Performance score, procedural im-plementation, knot quality, task time, and suture placement accuracy | Performance of last suture (practice trial) was assessed |

Giacomino et al. (2020), *PeerJ*, DOI 10.7717/peerj.10129

**Table 1** (*continued*)

| Study | Design/ Country | Participants | Procedure trained[a] | Teaching approach in experimental group | Teaching approach in control group | Time required for teaching | Student teacher ratio | Outcome measurements[a] | Endpoints |
|---|---|---|---|---|---|---|---|---|---|
| *Rossettini et al. (2017)* | RCT/ Italy | *n* = 39 third-year undergraduate physiotherapy students | Cervical C1- C2 spine mobilisation | Teaching using Peyton's four-step approach (demonstration, deconstruction, comprehension, performance) | "See one, do one" approach as reported by *Herrmann-Werner et al. (2013)* | Time required for teaching did not significantly differ between groups. | 3:1 | Performance checklist, time to teach; time to perform and student satisfaction | Post-test (after training), retention tests (1 week and 1 month after training) |
| *Ruesseler et al. (2019)* | Randomised controlled cohort study with 4-arms (the arms "video 4-step approach" and "See One - Do One") were included)/ Germany | *n* = 73 fourth-year medical students | Six procedures including three basic surgical skills ( replacement of a complex wound dressing, sterile covering, and performance of a suture) | Video 4-step approach: video supported step 1 and 2, the steps 3 and 4 were performed as reported by Peyton | "See one, do one", a trainer demonstrated the skill and explained. Followed by practice under supervision | Teaching units had equal duration between groups (day1: 90 min per unit; day 2–5: 210 min per unit) | max. 6:1 | OSCE with 6 stations, performance was rated on trinary checklist | Post-test (during training week) |

**Notes.**

[a]if multiple procedures or assessments were used in the primary studies the included procedures and assessments within this systematics review are underlined.

teaching (*Gradl-Dietsch et al., 2019*; *Gradl-Dietsch et al., 2018*; *Herrmann-Werner et al., 2013*; *Münster et al., 2016*), 2-stage teaching approach (*Jenko, Frangež & Manohin, 2012*), Orde's 2-step method (*Lapucci et al., 2018*; *Orde, Celenza & Pinder, 2010*), standard instructions (*Gradl-Dietsch et al., 2016*; *Krautter et al., 2011*), traditional bedside teaching (*Lund et al., 2012*) or see one, do one (*Rossettini et al., 2017*; *Ruesseler et al., 2019*). The time allocated to the teaching of the procedural skills was set equal in most included studies. Four studies (*Herrmann-Werner et al., 2013*; *Krautter et al., 2011*; *Lund et al., 2012*; *Rossettini et al., 2017*) used this variable as outcome measure. All of them reported that between the groups the same or a similar amount of time was required for teaching.

Data to evaluate the following comparisons were available:

- Peyton's teaching approach versus a standard teaching approach (PEY vs ST)
- Peyton's teaching approach with peer teaching versus a standard teaching approach with peer teaching (PeerPey vs PeerSt)
- Best practice skills lab with peer teaching versus a standard teaching approach with peer teaching (PeerBpsl vs PeerSt)
- Media supported Peyton's teaching approach versus a standard teaching approach (MPey-St)
- All forms of Peyton's teaching approach versus a standard teaching approach

During the controlling of the data set (https://doi.org/10.6084/m9.figshare.12619151) 7 data entries were flagged and double checked. This corresponded to 2.43% of the data set.

## Analysis of effectiveness

Below the analysis of effectiveness is presented reporting on two outcomes (i.e., performance and time needed to perform the procedure) at two different endpoints (i.e., after acquisition and after a retention period).

### *Performance—post-acquisition test*

Fourteen studies reporting on 17 samples with a total of 970 participants allocated to Peyton's teaching approach and 935 allocated to a standard teaching approach were included for the analysis of the outcome performance at post-acquisition testing. Four different sub-groups were identified. First, 9 studies compared Peyton's teaching approach against a standard teaching approach (*Balafoutas et al., 2019*; *Gradl-Dietsch et al., 2016*; *Jenko, Frangež & Manohin, 2012*; *Krautter et al., 2011*; *Lapucci et al., 2018*; *Lund et al., 2012*; *Orde, Celenza & Pinder, 2010*; *Romero et al., 2018*; *Rossettini et al., 2017*). The analysis showed an effect size of 0.5 SMD (95% CI [0.13–0.87]) in favour of the Peyton group. Heterogeneity was substantial with an $I^2$ of 69%. Three studies compared peer or student tutor Peyton's teaching versus peer standard teaching (*Gradl-Dietsch et al., 2019*; *Gradl-Dietsch et al., 2018*; *Münster et al., 2016*). The effect size was in favour of peer standard teaching with a SMD of −0.15 (95% CI between −0.23 and −0.06). Heterogeneity was not important within this comparison ($I^2$: 0%). One study reported on the comparison best practice skills lab (Peyton's teaching approach was part of the intervention) with peer tutors versus standard peer teaching (*Herrmann-Werner et al., 2013*). A large effect in favour of best practice skills lab training was identified (SMD: 1.38; 95% CI between

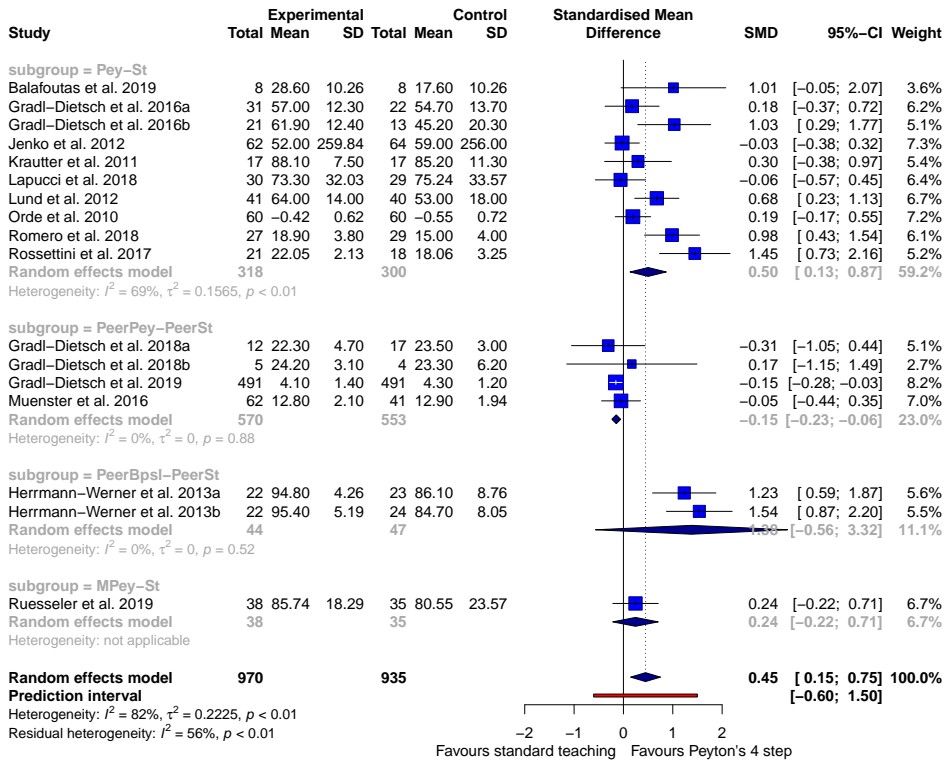

**Figure 2** **Forest plot performance - Peyton's 4-step versus standard teaching at post-acquisition testing.** Pey, Peyton's teaching; St, standard teaching; PeerPey, peer Peyton's teaching; PeerSt, peer standard teaching; PeerBpsl, peer best practice skills lab; MPey, Media supported Peyton. **NB**. *Gradl-Dietsch et al. (2018)* and *Gradl-Dietsch et al. (2016)* are presented as two samples because data for women and men are analysed separately (a: woman, b: men). Data from *Herrmann-Werner et al. (2013)* are presented as two samples (a: participants with a 3 months follow up, b: participants with a 6 months follow up).

−0.56 and 3.32). The I² was 0% for this analysis. The last subgroup compared a media supported Peyton's teaching approach versus standard teaching (*Ruesseler et al., 2019*). A small effect was analysed in favour of the Peyton group with a SMD of 0.24 and a 95% CI between −0.22 and 0.71. The overall model showed a small to moderate effect size in favour of Peyton's teaching approach with an effect size of 0.45 SMD (95%CI between 0.15 and 0.75). Heterogeneity was substantial with an I² value of 82%. A prediction interval between −0.6 and 1.5 was analysed (Fig. 2).

### Performance - retention test

Five studies were included for the outcome performance at retention testing. The studies reported a total of 169 participants in the Peyton group and 135 in the standard teaching group (Fig. 3).

It was possible to analyse three different subgroups. First, three studies reported on the comparison Peyton versus standard teaching (*Gradl-Dietsch et al., 2016*; *Orde, Celenza & Pinder, 2010*; *Rossettini et al., 2017*). A small to moderate effect in favour of the Peyton group was identified (SMD: 0.38; with a 95%CI between −0.14 and 0.9). Moderate

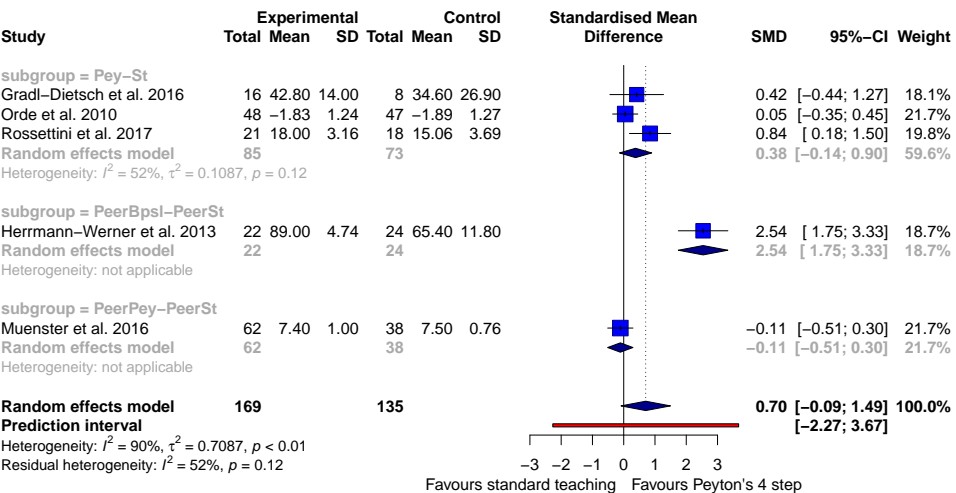

**Figure 3** Forest plot performance—Peyton's 4-step versus standard teaching at retention testing. PeerPey, peer Peyton's teaching; PeerSt, peer standard teaching; Pey, Peyton's teaching; St, standard teaching.

heterogeneity was analysed ($I^2$: 52%). The second subgroup compared peer best practice skills lab teaching with standard peer teaching (*Herrmann-Werner et al., 2013*). A large effect size was analysed in favour of best practice skills lab training SMD: 2.54 (95%CI between 1.75 and 3.33). The third subgroup compared Peyton's peer teaching with standard peer teaching. An SMD of −0.11 with a 95% CI between −0.51 and 0.3 in favour of peer standard teaching was analysed.

The random effects model over all subgroups showed a moderate to large effect size in favour of Peyton's teaching approach at retention testing (SMD: 0.7 with a 95%CI between −0.09 and 1.49). The heterogeneity of this analysis was large ($I^2$: 90%). The retention period ranged between 1 month (*Rossettini et al., 2017*) and 6 months (*Gradl-Dietsch et al., 2016*).

### Time needed for procedure—post-acquisition test

Six studies with a total of 657 participants in the Peyton group and 655 in the standard teaching group were included in this analysis (Fig. 4). Two different subgroups were identified. One study compared peer Peyton's teaching versus peer standard teaching (*Gradl-Dietsch et al., 2019*). An effect size of 0.05 SMD (95% CI between −0.07 and 0.18) was analysed. The second subgroup compared Peyton's teaching approach with standard teaching. Five studies were included in this analysis (*Krautter et al., 2011*; *Lund et al., 2012*; *Orde, Celenza & Pinder, 2010*; *Romero et al., 2018*; *Rossettini et al., 2017*). Findings were in favour of Peyton's teaching approach with a large effect size of −1.06 SMD and a 95% CI between −2.77 and 0.65. The overall model showed that participants in the Peyton groups needed considerably less time to perform the procedures at post-acquisition testing. A large effect size of −0.8 SMD (95%CI between −2.13 and 0.53) was associated with this finding.

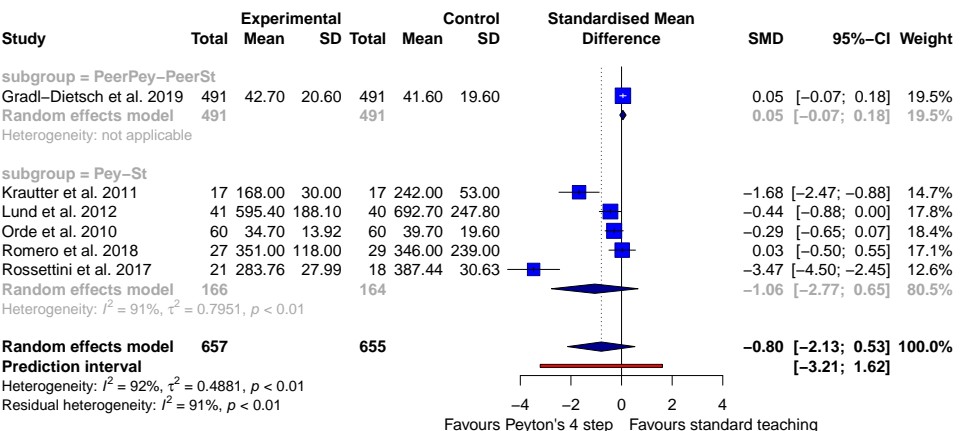

**Figure 4** **Forest plot time needed for procedure—Peyton's 4-step versus standard teaching at post-acquisition testing.** PeerPey, peer Peyton's teaching; PeerSt, peer standard teaching; Pey, Peyton's teaching; St, standard teaching.

The heterogeneity for this analysis was large with an $I^2$ of 92%. The prediction interval was between −3.21 and 1.62.

### Time needed for procedure—retention test

For the analysis time needed for the procedure at retention testing two studies were included (*Orde, Celenza & Pinder, 2010*; *Rossettini et al., 2017*). Both studies compared Peyton's 4 step teaching approach with a standard teaching approach. A large effect size of −2.65 SMD (95% CI[−7.77–2.47]) showed that the time needed to perform the procedure was considerable shorter after a training using Peyton's teaching approach. Heterogeneity was large ($I^2$: 98%). The retention period ranged between 1 month (*Rossettini et al., 2017*) and 2 months (*Orde, Celenza & Pinder, 2010*).

### Meta-regression student teacher-ratio—performance post-acquisition

A univariable meta-regression was performed to analyse whether the student-teacher ratio was an independent predictor of performance on post-acquisition tests. All studies from the meta-analysis "performance - post-acquisition test" with exception of the study of Ordre et al. (2010) (i.e., the authors did not report the student-teacher ratio) were included into the meta-regression. The meta-regression showed that the effectiveness of Peyton's teaching approach was higher in studies with fewer of students per teacher (Fig. 5). The overall model explained 57% of the variability of the effect sizes (p: 0.01, r2: 56.86%) and the students per teacher variable showed that for one student more per teacher, the effect size was reduced by 0.08. This association was statistically significant (b1: −0.08 (95% CI [−0.14 to −0.0232]), t: −2.96, p: 0.01).

## Risk of bias

The risk of bias was low for all studies regarding the item random sequence generation with exception of the study of Ruesseler and colleagues (*2019*), which was classified as unclear. Regarding the allocation concealment most studies were rated as unclear with exception

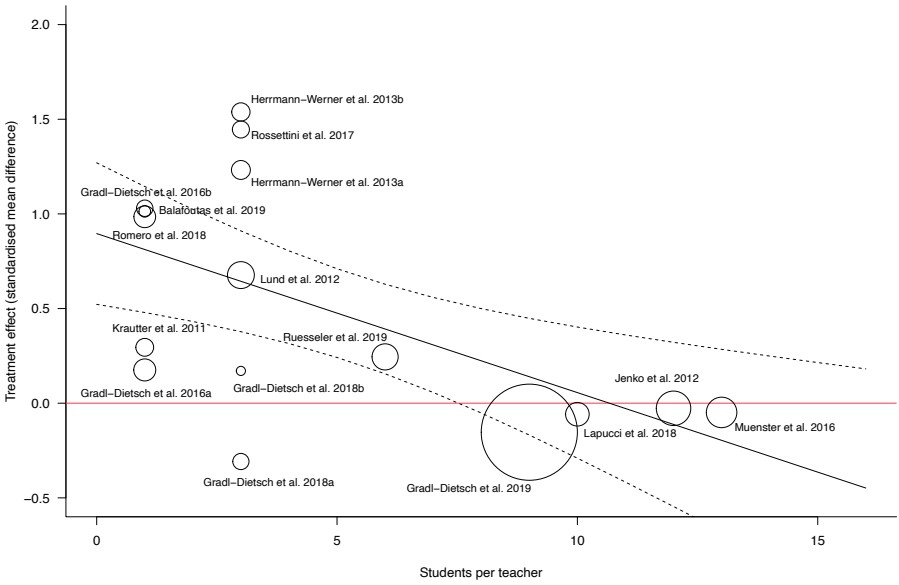

**Figure 5** **Scatterplot meta-regression students per teacher as predictor for performance at post-acquisition testing.** The red line represents the line of equal effectiveness between Peyton's teaching approach and standard teaching. The predicted regression line is plotted in black with corresponding confidence intervals.

of two studies (*Gradl-Dietsch et al., 2019*; *Jenko, Frangež & Manohin, 2012*). Blinding of participants and personnel was rated as high risk of bias in all studies with exception of the study of *Rossettini et al. (2017)*.

The authors stated that the participants and teachers were blinded to the aims of the study. The risk of bias regarding outcome assessment was low. Only two studies were rated as unclear regarding this risk of bias item blinding of outcome assessment (*Lapucci et al., 2018*; *Münster et al., 2016*). One study was assessed as having a high risk of bias regarding incomplete outcome assessment because a relatively high number of study discontinuations were reported (*Münster et al., 2016*). A summary risk of bias plot is presented in Fig. 6. Regarding the agreement of the human reviewer and the machine learning algorithm it was possible to compare 48 risk of bias evaluations. No conflicts occurred in 37 (77%) decisions and 11 (23%) decisions resulted in a conflict.

## Sensitivity analyses

Findings from a crossover study of Gradl-Dietsch and co-workers (*2019*) were integrated into the meta-analysis and the study was treated as parallel group trial. In order to address a potential unit of analysis issue a sensitivity analysis was performed. Because data from paired analyses were not available we adjusted the study data based on a method described by *Elbourne et al. (2002)*. A correlation coefficient derived from the data of *Lund et al. (2012)* was used to calculate an adjusted standard error.

For the meta-analysis performance at post-acquisition, the standard error of the study decreased from 0.06 to 0.04. The effect estimate of the analysis peer Peyton versus peer
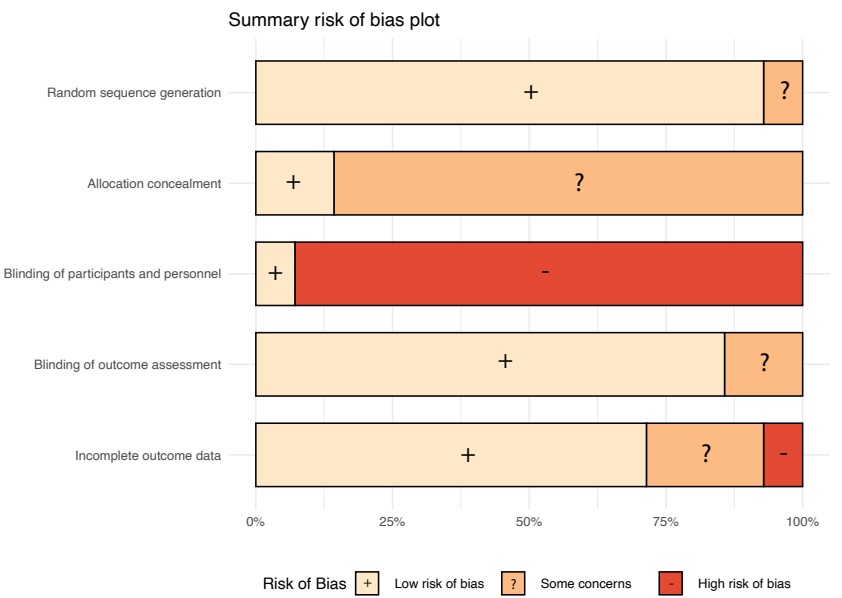

**Figure 6   Summary risk of bias plot.**

standard teaching remained $-0.15$ SMD with a slightly changed 95% CI between $-0.22$ to $-0.08$.

The adjusted standard error had only minimal influence on the meta-regression of the student teacher ratio at post-acquisition. The overall model (p: 0.01, r2: 57.54%) and the students per teacher variable (b1: $-0.08$ (95% CI[$-0.14$ to $-0.0232$]), t: $-2.96$, p: 0.01) remained significantly related to the mean effect size.

Within the meta-analysis time needed for the procedure at post-acquisition testing the sensitivity analysis resulted in a slightly smaller standard error of the *Gradl-Dietsch et al. (2019)* study. Therefore, the effect estimate of the comparison peer Peyton's teaching versus peer standard teaching changed to 0.05 SMD with a 95% CI between -0.05 and 0.16. The effect estimate of the overall model did not change.

## DISCUSSION

This systematic review with meta-analysis and integrated meta-regression set out to evaluate the effectiveness of Peyton's teaching approach compared with a standard teaching approach. The primary finding was that Peyton's teaching approach was more effective than a standard teaching approach on the acquisition of procedural skills at post-acquisition testing. A small to moderate effect size was associated with this finding. However, different subgroups of Peytons's teaching approach were analysed and effectiveness differed between subgroups. Two comparisons showed findings in favour of Peyton's teaching approach when the procedure was instructed by teachers or faculty members (i.e., Peyton versus standard teaching and media supported Peyton's teaching approach versus a standard teaching approach). Two comparisons used peers to perform the procedural skills training. Peer Peyton versus peer standard teaching showed inconclusive results with a small effect

size in favour of peer standard teaching. In contrast the comparison peer best practice skills lab versus peer standard teaching showed a large effect size in favour of peer best practice skills lab. Therefore, it remains unclear whether Peyton's teaching approach is effective when peers are used as tutors for the outcome skill acquisition.

The meta-analysis of skill acquisition at retention testing was in favour of Peyton's teaching approach with a moderate to large effect size. Both subgroups were in favour of Peyton's approach. However, the effect size for the experimental group was considerable smaller compared to the findings at post-acquisition testing. The comparison peer best practice skills lab versus peer standard teaching showed a large effect size. Considerable larger than the effect size at post-acquisition testing. However, only one study reported on this comparison and more studies are needed to confirm this finding.

Regarding the outcome time needed to perform the procedure the findings indicated that participants needed considerably less time to perform a procedure if Peyton's teaching approach was instructed by teachers or faculty members. One study showed a very large effect (*Rossettini et al., 2017*). This study showed some educational differences to the other studies in the analysis. For example, participants from physiotherapy education were used and the trained procedure was a cervical spine mobilisation. In addition, relatively few students per teacher participated in the teaching events. The potential influence of the different procedures on the effect estimate should be investigated in future studies.

An increased effectiveness of Peyton's teaching approach at retention testing was analysed. This was mainly seen in the time needed for procedure outcome. The possible long-term comprehension advantage of Peyton's teaching approach has been previously discussed by *Herrmann-Werner et al. (2013)*. The authors showed that Peyton's teaching approach had an increased long-term effect on the acquisition of simple and complex skills. This finding is of educational importance because deterioration of procedural skills is likely after several weeks (*Bonrath et al., 2012*) and Peyton's teaching approach could be a useful educational method to reduce this.

The meta-regression with the student-teacher ratio as independent predictor showed that Peyton's teaching approach was more effective in groups with fewer students per teacher. This supports the idea that Peyton's teaching approach was designed for a teaching ratio of 1:1 (*Nikendei et al., 2014*). The student-teacher ratio of the analysed studies ranged between 13:1 (*Münster et al., 2016*) and several studies using a 1:1 ratio (*Balafoutas et al., 2019*; *Gradl-Dietsch et al., 2016*; *Krautter et al., 2011*; *Romero et al., 2018*). In studies where 9 or more students per teacher were used the treatment effect was close to zero. The highest effect sizes were analysed in studies using a student teacher ratio of 3:1 (*Herrmann-Werner et al., 2013*; *Rossettini et al., 2017*). This indicates that Peyton's teaching approach should ideally be used in groups with 1 to 3 students per teacher. If this is not possible, it could be argued that group sizes with less than 9 students per teacher are still in favour of Peyton's teaching approach.

Furthermore, it should be reported that *Münster et al. (2016)* reported a median group size of 13 students with a range between 9 and 13 participants and *Ruesseler et al. (2019)* reported a maximum group size of 6 participants per teacher. These summary estimates of the variable were used within the meta-regression, but this might have caused some

imprecision. In addition, the variable student-teacher ratio was not reported in the study of *Orde, Celenza & Pinder (2010)* and therefore the study was not included into the meta-regression.

The control intervention in this review was labelled as "standard teaching" approach. However, the educational approaches used within the control arms presented a source of heterogeneity. A broad range of approaches was identified such as: Halsted teaching, 2-stage teaching approach, Orde's 2-step method, standard instructions, traditional bedside teaching or see one - do one. These educational approaches show considerable similarities but are not exactly the same interventions. However, all of the standard teaching approaches have in common that they did not include the third step of Peyton's teaching approach (i.e., guiding the teacher through the procedure), which is assumed to be beneficial for skill acquisition (*Gradl-Dietsch et al., 2016*; *Rossettini et al., 2017*). To deal with these differences several subgroup analyses were performed. In addition, the meta-analysis was performed using a random effects model. Within the subgroups the statistical heterogeneity was considerable smaller compared to the overall analyses. The overall analyses showed substantial heterogeneity and should therefore be analysed with caution.

Eligible outcome assessments for this systematic review were assessments of procedural skills, which could be a procedure specific checklist or a global rating scale. However, when studies reported both types of assessments, the checklists were preferred. This was justified on the basis of the suggested best methods for evaluation by the Accreditation Council for Graduate Medical Education (ACGME) (*ACGME, 2000*; *Swing, 2002*). Within the guideline, checklists are recommended as "most desirable" when assessing medical procedures. Rating scales are recommended as "potentially applicable method". Therefore, we preferred data based on procedure specific checklists. However, this is a controversial topic and some authors have reported that global rating scales have additional values and should be used when procedural skills are evaluated (*Ma et al., 2012*; *Regehr et al., 1998*).

## Limitations

Several other potential effect modifiers exist, which were not explored in this study because we did not specify these analyses in the study protocol. First, *Gradl-Dietsch et al. (2016)* reported that gender might be considered as potential moderator variable for the effectiveness of Peyton's teaching approach. Within their study the authors suggested that men might benefit more from Peyton's teaching approach compared to women. This could be explained by the results of *Ali et al. (2015)*. The authors reported in a systematic review that the acquisition of surgical skills differs between men and women. However, it is difficult to investigate the gender variable with a meta-regression because relatively few studies reported the findings for men and women separately.

Second, acquiring simple procedures is different from acquiring complex skills (*Wulf & Shea, 2002*). Therefore, the complexity of the procedural skills might affect the effectiveness of Peyton's teaching approach. However, rating the complexity of the included procedures is challenging as procedures from various domains of health professions education were included. Third, the experience of the teacher teaching the procedural skill and the

experience of the students learning the skill might affect the effectiveness of Peyton's teaching approach.

Findings from a crossover trial of Gradl-Dietsch and co-workers (*2019*) were integrated into the meta-analysis. Findings from a paired analysis were not available and therefore we used the reported values and treated the study as a parallel group trial.

However, when the results of randomised controlled trials and crossover studies are combined, the results of crossover studies should be based on paired analyses (*Elbourne et al., 2002*). If findings from unpaired analyses are used the confidence intervals are likely too wide and this might give rise to a unit of analysis issue (*Higgins et al., 2019*). As a consequence, we performed a sensitivity analysis and adjusted the standard errors using a method described by *Elbourne et al. (2002)*. A correlation coefficient derived from the data of *Lund et al. (2012)* was used to calculate the adjusted standard errors. Unfortunately, it was only possible to calculate the correlation coefficient using the Lund et al. study. The remaining studies did not provide sufficient data. However, findings remained similar after the sensitivity analysis. The only differences were slightly changed 95% confidence intervals. We have therefore decided to include the study by *Gradl-Dietsch et al. (2019)* in the analysis.

An additional limitation of this review might be that we did not include studies reporting about the effectiveness of George and Doto's teaching approach (2001). Peyton's and George and Doto's teaching approach are similar regarding their stepwise teaching structure. However, the inclusion of this additional educational intervention would have increased the heterogeneity considerably. In view of the relatively high proportion of analysed heterogeneity within our pairwise analyses, we decided against it. However, in the context of a network meta-analysis future studies could possibly compare these two and other reported teaching approaches for the acquisition of procedural skills.

## Implications for research

Several implications for research were identified. First, the effectiveness of Peyton's teaching approach on skill acquisition should be explored in various health professions. The included studies reported on the use of Peyton's teaching approach in medical education. Only three studies were found analysing this approach in other health professions. Further studies are therefore needed to investigate this approach in the field of nursing or physiotherapy. Second, the proposed moderator variables gender, skill complexity and level of experience of teacher and students should be further explored. Third, more evidence is needed regarding the use of peer teachers. Fourth, the high effectiveness of the best practice skills lab training should be explored in further studies. In addition, future studies should investigate a stabilised learning of motor skills with long-term follow up (during the retention phase). Moreover, there is a need to consider also the assessment of the motor skill acquired in ecological settings (e.g., during internships) suggesting an adequate transfer phase.

## Implications for practice

Peyton's teaching approach is effective for the acquisition of procedural skills. The evidence is robust for the field of medical education. One might assume that the acquisition of

skills in other health professions could also benefit from Peyton's teaching approach. However, this must be further investigated. When Peyton's teaching approach is used the number of students per teacher should be small (e.g., ranging between 1 and 3 students per teacher) to be more effective than a standard teaching approach. Implications for teachers in different healthcare fields (e.g., nursing, physiotherapy or speech and language therapy education) are less robust. However, some procedures within this review are used across healthcare fields. For example, procedures from manual therapy were used in medical education (*Gradl-Dietsch et al., 2016*) and in physiotherapy education (*Rossettini et al., 2017*). Educators teaching these procedural skills in different healthcare fields are encouraged to use Peyton's teaching approach (i.e., within the discussed limitations). In addition, given the broad spectrum of included procedures in this review it seems likely that Peyton's teaching approach also applies to procedures in different healthcare fields, but this needs further investigation.

## CONCLUSIONS

Peyton's teaching approach is an effective teaching approach for skill acquisition of procedural skills when faculty members are used as teachers. When peer students or student tutors are used as teachers the effectiveness of Peyton's teaching approach is less clear. Peyton's teaching approach is more effective when small groups with few students per teacher are used.

### List of abbreviations

| | |
|---|---|
| **MPey** | Media supported Peyton |
| **PeerBpsl** | peer best practice skills lab |
| **PeerPey** | peer Peyton's teaching |
| **PeerSt** | peer standard teaching |
| **Pey** | Peyton's teaching |
| **St** | standard teaching |

### Funding
The authors received no funding for this work.

### Competing Interests
The authors declare there are no competing interests.

### Author Contributions
- Katia Giacomino performed the experiments, analyzed the data, authored or reviewed drafts of the paper, and approved the final draft.
- Rahel Caliesch performed the experiments, authored or reviewed drafts of the paper, and approved the final draft.
- Karl Martin Sattelmayer conceived and designed the experiments, performed the experiments, analyzed the data, prepared figures and/or tables, authored or reviewed drafts of the paper, and approved the final draft.

## Data Availability

The data set is available at Figshare:

Sattelmayer, Martin (2020): Data_SR_Peyton.xlsx. figshare. Dataset. https://doi.org/10.6084/m9.figshare.12619151.v1.

## Supplemental Information

Supplemental information for this article can be found online at http://dx.doi.org/10.7717/peerj.10129#supplemental-information.

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
