# Peer review of "The effectiveness of the Peyton’s 4-step teaching approach on skill acquisition of procedures in health professions education: A systematic review and meta-analysis with integrated meta-regression"

_PeerJ, doi:10.7717/peerj.10129_

## Round 0.1 · original submission · Minor Revisions

Please address the minor issues point out by the reviewers, in particular those aimed at expanding your background information and streghtening your discussion. This will significantly straighten the manuscript before it is ready for publication.

·

Basic reporting

The manuscript is clear and unambiguous. Professional English has been used throughout the manuscript, the abstract is informative, there is a clear description of the methods and the results are reported accurately. The structure of the introduction and the discussion are adequate.
The bibliography is up to date and the studies, which are mentioned in table 1, are depicted correctly with all the important characteristics that are relevant for this review. The figures are in good quality and the plots are self-explaining. The results are relevant to the hypotheses. An important strength of the review is that the authors managed to deal with the heterogeneity of the available studies through an exact categorization of the outcomes and, importantly, through definition of the comparison groups.

Experimental design

The review is within the scope of the journal, as it refers to both medical and health sciences from the educational point of view. The endpoints are meaningful: Proof on the effectiveness of the Peyton’s approach as an educational principle can facilitate better procedural skills of trainees through modification of institutional teaching standards. Proof on the importance of the low student-to-teacher ratio can also contribute to teaching that is more effective. The methods are well described.

Validity of the findings

There are numerous reports on the superiority of the Peyton’s teaching approach in the acquisition of specific skills, however this review demonstrates that the method can be generally adapted in cases of experienced tutors. Additionally, the limitations in case of peer-teachers are clearly described. The underlying data are available. The results completely support the conclusions

Additional comments

Some minor issues:

Abstract: In the “Background” section both aims of the study are defined. In Line 266 the effectiveness is reported on two outcomes. The authors may choose to include the time-needed-for-procedure variable in the “Results” section of the abstract.

In the introduction (lines 100-101) the time-for-teaching variable is referred. A comment in the included studies section of the results may state the fact that the time-for-teaching was identical in both groups (i.e. 3h in Grandl-Dietsch 2019, 90 min in Grandl-Dietsch 2016 …).

In the Introduction (lines 63-65) a definition is given for the procedural skills. This definition could be omitted, since it neither has an impact in the interpretation of the results, nor is safety an endpoint in the participating studies (only the “correct” performance of the procedure). Alternatively, the authors could state that the procedural skills relate to the parameters examined by each author in his/her field of expertise.

(Line 68-69) Sentence is missing the verb

Line 184: Hedges’ g is by most authors capital

In the discussion there should be a comment on the increased effect of Peyton’s approach in the retention testing (vs post-acquisition testing), mainly in the time-needed-for-procedure variable. The possible long-term comprehension advantage of deconstructive teaching has been previously discussed (see references).

·

Basic reporting

The manuscript is well written and is a pleasant read. However, as I have stated in my comments to the author, the manuscript will have a greater outreach if it is published in a medical education journal. These include:
Academic Medicine
BMC Medical Education (Open access)
Medical Teacher etc.

Experimental design

The authors have described the design suitably.

Validity of the findings

The manuscript adds to the corpus of literature with regards to instructional design strategies employed in the delivery of the curriculum in Undergraduate and Graduate medical education.

Additional comments

This is a very well designed and well written manuscript, which was indeed a pleasant read. The article is an important addition to the corpus of literature on instructional design strategy. I do not have any major concerns to state.
The following are some aspects, which I believe will further strengthen the study, which are in fact minor edits to be included. Further to the above, I also believe that this article will be more suitable for publication in a health professions education journal such as Academic Medicine, BMC Medical Education, Medical Teacher etc. However, this is a minor point for the authors to consider.

1. The authors have rightly discussed the advantages of adopting Peyton’s 4-step approach over traditional pedagogical strategy in the dissemination of complex procedural skills in medical education. However, it would add to the manuscript if they can also touch upon George and Doto's 5-step approach.

2. Yes, the third step of Peyton is the crucial step, but so is the fourth, as in this step (Elicit performance), the instructor provides feedback to the learner. In fact, a systematic review by Issenberg et al a key component of clinical skill teaching and learning are the opportunity for feedback while practicing the skill (the authors are requested to refer to Features and uses of high-fidelity medical simulations that lead to effective learning: a BEME systematic review, Medical Teacher, 27 (1) (2005), pp. 10 -28 f0r details). Additionally, this step of Peyton’s pedagogical strategy is also supported by Bandura’s scaffolding theory (refer to Learning theories by DH Schunk) . The above aspects should be included in the section of the introduction where the authors have elaborated on the individual steps.

3. One of the strengths of Peyton instructional design strategy is that it can be effectively combined with other instructional design strategies, which allows the simultaneous delivery of theoretical concepts along with complex procedural skills. In fact, this reviewer hasn’t come across any other pedagogical strategy except Peyton, which has been effectively employed in the design of a blended pedagogical approach. This aspect should be included in the introduction which will strengthen the rationale of the study. (The authors can refer to the following articles: Blending Gagne's Instructional Model with Peyton's Approach to Design an Introductory Bioinformatics Lesson Plan for Medical Students: Proof-of-Concept Study. JMIR Med Educ. 2018;4(2):e11122.; Combining Peyton's four-step approach and Gagne's instructional model in teaching slit-lamp examination.” Perspectives on medical education vol. 3,6 (2014): 480-5.)

Other sections of the manuscript are well written and I have nothing significant to add to them

Reviewer 3 ·

Basic reporting

no comment

Experimental design

no comment

Validity of the findings

no comment

Additional comments

Dear Authors
Thanks a lot for the opportunity you have offered me to revise this interesting systematic review with meta-analysis.
Globally, the manuscript is well conducted and the reporting adheres to the PRISMA guideline. There are only minor revisions that need to be addressed before the acceptance.
I wait your revised version of the manuscript.
Thanks a lot for your effort in preparing this interesting paper.
Best regards.

Material and methods:
-Searches: please could you report who performed the search phase? Moreover, could you explode the acronym ERIC as “Education Resources Information Center”
-line 130: please consider to change “was build” in “was prepared”.
-Study selection, data extractions, risk of bias assessment: please could you report how you have measure the disagreement among authors, thus including a statistical measure (e.g., k, ICC? With 95%IC)

Risk of bias:
-line 362-363: I have checked the original paper of Rossettini et al. The authors declared in the abstract: “Participants, teachers and assessors were blinded to the aims of the study”. Moreover, they claim in the material and methods “Teachers were blinded to the study outcomes”, “Each teacher was trained in only one method in order to prevent contamination during the sessions”, and “Both teachers received a detailed sheet regarding students’ learning goals, the duration of the session and the steps required by the approach“. Accordingly, please modify your sentence aimed to be adherent with the original manuscript.

Discussion
-line 455: please could you explode the acronym “ACGME”?

Implications for research:
-I suggest authors to add also another implication regard learning (as suggested by Wulf’s research group). We consider an “acquisition phase” when we acquire a motor skill at short term; while a “retention” when there is a follow up and a “transfer phase” when we translate the learning in other ecological settings. Accordingly, future studies should investigate a stabilized learning of motor skills with long-term follow up (during retention phase). Moreover, there is a need to consider also the assessment of the motor skill acquired in ecological settings (e.g., during internship) suggesting an adequate transfer phase.

Conclusion:
The findings of this study are really relevant for educators. I suggest authors also to include other possible implications for inter professional education. Accordingly, how educators involved in different healthcare fields (e.g., nursing, medicine, physiotherapy, speech therapy) can implement your findings in their educational practice?

Appendix 1:
I thank you for providing the Appendix 1, however please add also the search strategies for all the databases. This strategy will improve the overall transparency of the manuscript.

References.
In the introduction section please include these fundamental references when you introduce the Peyton’s four approach.
* American College of Surgeons. Advanced Trauma Life Support for Doctors, 6th edn. Chicago, IL: American College of Surgeons 1997.
* George JH, Doto FX. A simple five-step method for teaching clinical skills. Fam Med 2001;33 (8):577–8.

---

## Round 0.2 · accepted · Accept

Thank you for addressing all the reviewer's comments. Your edits have improved the quality of the manuscript I can now recommend it for publication. There are a few minor comments from the reviewer that need addressing, please do so before or during proofing.

Reviewer 3 ·

Basic reporting

Please revise some typo
sometimes you use e.g., other times i.e.,
line 64 psychmotor

Experimental design

No comment

Validity of the findings

No comment

Additional comments

Dear Authors
congratulations for your hard work.
I suggest the editor to accept the paper.
I'm waiting to read the published paper.
Best regards.